# Intravesical Therapy for Upper Urinary Tract Urothelial Carcinoma: A Comprehensive Review

**DOI:** 10.3390/cancers15205020

**Published:** 2023-10-17

**Authors:** Zheng Wang, Haoqing Shi, Yifan Xu, Yu Fang, Jiaao Song, Wentao Jiang, Demeng Xia, Zhenjie Wu, Linhui Wang

**Affiliations:** 1Department of Urology, Changhai Hospital, Naval Medical University, Shanghai 200433, China; wangzheng@smmu.edu.cn (Z.W.); shihaoqing2007@163.com (H.S.); xuyifan@smmu.edu.cn (Y.X.); 13916943129@163.com (Y.F.); songjiaao@smmu.edu.cn (J.S.); jiangwentao@smmu.edu.cn (W.J.); 2Department of Pharmacy, Seventh People’s Hospital of Shanghai University of Traditional Chinese Medicine, Shanghai 200137, China; demengxia@smmu.edu.cn

**Keywords:** upper urinary tract urothelial carcinoma, intravesical recurrence, intravesical therapy, bladder instillation, chemotherapy, ureteroscopy, radical nephroureterectomy, UGN-101, mitomycin C, Bacillus Calmette–Guerin

## Abstract

**Simple Summary:**

This comprehensive review discusses the current status and future prospects of intravesical therapy for upper urinary tract urothelial carcinoma (UTUC). It emphasizes the need to understand its role in UTUC management and the importance of personalized strategies for drug selection, dosage, timing and frequency to optimize treatment outcomes and reduce intravesical recurrence. By summarizing historical development, clinical trials, guideline recommendations, and clinical applications, this review provides valuable insights. We aim to guide future studies and impact the research in UTUC, advancing the understanding and utilization of intravesical therapy for UTUC.

**Abstract:**

Upper tract urothelial carcinoma (UTUC) poses unique challenges in diagnosis and treatment. This comprehensive review focuses on prophylactic intravesical therapy for UTUC, summarizing key aspects of intravesical therapy in various clinical scenarios, including concurrent with or following radical nephroureterectomy, kidney-sparing surgery, ureteroscopy-guided biopsy. The incidence of intravesical recurrence in UTUC after surgical treatment is significant, necessitating effective preventive measures. Intravesical therapy plays a vital role in reducing the risk of bladder recurrence following UTUC surgery. Tailoring timing, drug selection, dosage, and frequency is vital in optimizing treatment outcomes and reducing intravesical recurrence risk in UTUC. This review provides a comprehensive summary of the history, clinical trials, guideline recommendations, and clinical applications of intravesical therapy for UTUC. It also discusses the future directions based on current clinical needs and ongoing trials. Future directions entail optimizing dosage, treatment duration, and drug selection, as well as exploring novel agents and combination therapies. Intravesical therapy holds tremendous potential in improving outcomes for UTUC patients and reducing the risk of bladder recurrence. Although advancements have been made in UTUC treatment research, further refinements are necessary to enhance efficacy and safety.

## 1. Introduction

Upper tract urothelial carcinoma (UTUC) is a rare and challenging malignancy that primarily affects the inner urothelial lining of the renal pelvis, calyces, and ureters. It accounts for approximately 5–10% of all urothelial carcinoma (UC) and presents unique diagnostic and therapeutic considerations [1].

The incidence of UTUC is on the rise, with an estimated annual incidence of 1–2 cases per 100,000 individuals. Moreover, there is an observable trend of UTUC affecting patients at older ages, as reflected by the increasing mean age at diagnosis from 71.5 to 73.4 years. Encouragingly, advancements in treatment strategies have contributed to improved prognoses for UTUC patients, as demonstrated by an upward trend in the five-year cancer-specific survival rate from 57.4% to 65.4% [2,3].

After treatment, the intravesical recurrence (IVR) of UTUC is observed in 22–47% of patients, with the rate varying based on the initial tumor grade [4,5,6,7]. IVR of UTUC is believed to occur through a dual-stage process known as the “seeding” and “field” hypothesis [8,9,10,11]. In this theory, short-term recurrences primarily occur due to the dissemination of tumor cells, while long-term recurrences are associated with the field effect, indicating the presence of a molecularly altered urothelium that is prone to tumor development. A shorter interval between IVR and UTUC is considered indicative of a poorer prognosis [12]. Moreover, the period of 2–2.5 years post-surgery is considered a high-risk phase for IVR, highlighting the importance of implementing adjuvant therapies in conjunction with surgery to reduce IVR [13,14,15,16,17]. This aspect not only warrants significant attention but also necessitates further research and investigation.

Although UTUC and bladder carcinomas share common pathogenic mechanisms and exhibit similar tumor characteristics, leading to the adoption of bladder cancer treatment strategies as a reference point for UTUC management [18,19], it is, however, important to note that UTUC and BC also have distinct differences, leading to variations in treatment approaches [20]. While significant progress has been made in UTUC treatment research, further advancements and refinements are still needed in comparison to the advancements made in non-muscle-invasive bladder cancer (NMIBC) [21].

Based on the American Urological Association (AUA) guidelines, it is strongly recommended that UTUC patients meeting the eligibility criteria should undergo a postoperative intravesical therapy to decrease the risk of IVR (Grade A evidence level) [8]. Currently, research on intravesical therapy for UTUC primarily focuses on postoperative bladder instillation. If a concurrent bladder tumor is present at the time of UTUC diagnosis or if there is bladder tumor recurrence after nephroureterectomy (RNU), treatment strategies for NMIBC can be considered as a reference.

Thus, the primary focus of this review is to comprehensively summarize and discuss the key elements pertaining to postoperative intravesical therapy for UTUC. These elements encompass strategy development, optimal timing and duration of therapy, drug selection, and emerging therapeutic agents. Through an in-depth analysis, this review aims to provide valuable insights into the field of intravesical therapy for UTUC in order to enhance the prevention of bladder recurrence and improve patient outcomes.

## 2. Definition of Intravesical Therapy

Intravesical therapy, also commonly referred to as bladder instillation, involves the administration of drugs or solutions directly into the bladder. This therapeutic approach aims to prevent or treat tumor recurrence within the bladder following UTUC surgery. By delivering medications directly to the bladder lining, intravesical therapy targets residual cancer cells and inhibits their growth, leading to improved treatment outcomes. It is an important adjunctive treatment option that requires careful monitoring and individualized selection of chemotherapy agents, immunotherapies, or other medications [6]. According to Hwang et al., a single-dose intravesical chemotherapy instillation significantly reduces bladder cancer recurrence risk compared to no instillation. During a 12-month follow-up period, prophylactic intravesical instillation could potentially lead to a significant reduction of 127 bladder cancer recurrences per 1000 participants [22].

The risk stratification influences the choice of treatment strategies for bladder management, and different surgical approaches are recommended for individuals with different risk categories. For patients with low-risk tumors, kidney-sparing management is offered as a preferred treatment option. In contrast, radical nephroureterectomy (RNU) (both open and minimally invasive), along with complete bladder cuff excision (BCE), is considered the standard surgical approach for localized high-risk UTUC [23,24,25,26,27]. Therefore, it can be concluded that patients undergoing different surgical procedures based on their risk profile would require distinct postoperative intravesical instillation treatment plans (Figure 1).

## 3. Search Strategy

A comprehensive literature search was conducted using the following databases: Pubmed, Web of Science, Medline, Embase, Cochrane controlled trials databases, and clinicaltrials.gov (accessed on 30 July 2023). Additionally, guidelines and abstracts from relevant associations and conferences, including the European Association of Urology (EAU), American Urological Association (AUA) and American Society of Clinical Oncology (ASCO), among others, were also reviewed. The search strategy employed the following keywords:

Intravesical therapy, intravesical treatment, intravesical instillation, bladder instillation, local therapy, adjuvant therapy, intravesical chemotherapy, upper urinary tract urothelial carcinoma, upper tract urothelial carcinoma, renal pelvis carcinoma, ureteral carcinoma, bladder recurrence.

The search strategy aimed to identify studies and the literature related to intravesical therapy for upper urinary tract urothelial carcinoma. The databases were searched for articles published up until the date of the literature search.

The search results were imported into reference management software (Endnote) for initial screening. Titles and abstracts were screened to identify potentially relevant articles. Full texts of the selected articles were retrieved and reviewed for eligibility based on predefined inclusion and exclusion criteria. Inclusion criteria encompassed studies that examined intravesical therapy for upper urinary tract urothelial carcinoma, including clinical trials, observational studies, and systematic reviews. Studies that merely focused on non-urothelial malignancies were excluded.

Data extraction was performed on the included studies, capturing relevant information such as study design, patient characteristics, intervention details, outcomes assessed, and key findings. The extracted data were synthesized and presented in a narrative format, highlighting the main findings, trends, and limitations of the included studies.

The quality of the included studies was assessed using appropriate tools based on the study design. For randomized controlled trials, the Cochrane Collaboration’s risk of bias tool was used, while observational studies were evaluated using relevant quality assessment tools.

The limitations of this review include potential publication bias and the exclusion of non-English articles, which may introduce a language bias. Efforts were made to minimize bias by conducting a comprehensive search across multiple databases and including diverse sources of evidence.

Overall, the search strategy outlined above aimed to identify relevant studies on intravesical therapy for upper urinary tract urothelial carcinoma from a variety of databases and sources. Clinical studies related to intravesical therapy are listed in Table 1, including completed trials, studies currently recruiting patients, trials about to commence recruitment, as well as clinical trials with an unknown status. The findings from this comprehensive review will contribute to a thorough understanding of the topic and provide valuable insights for clinical practice and future research.

## 4. Intraoperative Bladder Instillation and Irrigation

Intraoperative intravesical therapy primarily involves two approaches. One approach is the administration of medication directly during the surgical procedure, aiming to maximize therapeutic efficacy by avoiding delays caused by drug absorption through the mucosa. The other approach involves continuous bladder irrigation with saline or distilled water to prevent the potential dissemination of tumor cells from the surgical site to the bladder.

Continuous irrigation, predominantly utilizing non-medicated solutions such as distilled water or saline, serves as a mechanism to perpetually flush the bladder. This approach is grounded in the rationale that dislodged tumor cells are effectively eliminated. In this study investigating the effect of intraoperative irrigation, a total of 109 UTUC patients with a median follow-up of 26.1 months were included [30]. Among them, 48 patients received bladder irrigation with either normal saline or distilled water intraoperatively. In the irrigation group, the recurrence rate was significantly lower compared to the non-irrigation group, with rates of 25.0% vs. 52.5%, respectively (*p* = 0.0066).

The instillation of chemotherapeutic agents presents a more assertive strategy. Direct administration into the bladder is designed to annihilate free-floating tumor cells on contact. MMC (Mitomycin C) is a potent chemotherapeutic drug widely employed in the management of various malignancies [35]. When used in bladder instillation, MMC plays a crucial role in targeting and treating bladder cancer, thereby minimizing the risk of recurrence and enhancing overall patient prognosis. In a retrospective analysis, 30 patients who received intraoperative (IO) instillation of MMC during RNU were compared with 21 patients who received postoperative (PO) instillation for UTUC. The estimated probability of 1-year bladder tumor recurrence rates was 16% in the IO group and 33% in the PO group (*p* = 0.09). Cox analysis revealed a significantly lower rate of recurrence rate in the first year postoperatively in the IO group (HR = 0.113, 95% CI = 0.28–0.63, *p* = 0.01) [36]. However, this approach introduces challenges related to timing. Given the variability in surgical duration, determining the optimal drug retention period becomes critical. Late conclusion of surgery might necessitate reconsideration of drug retention to mitigate potential toxicities, as opposed to adhering to a standardized duration.

In the study conducted by Nadler et al., MMC (40 mg MMC in 40 mL 0.9% saline) was instilled in 47 patients after BCE and retained for a maximum duration of one hour during RNU [28]. The safety and feasibility of instillation during the surgical procedure have been confirmed, as no complications were observed. However, due to limited sample size and trial design, the efficacy of MMC in suppressing bladder recurrence has not been adequately validated.

Immunotherapy, with agents like Bacillus Calmette–Guerin (BCG), offers an alternative therapeutic mechanism. Rather than direct cytotoxicity, the goal is to harness the body’s immune response to target and obliterate tumor cells. The enduring effect of this method, even post-agent removal, underscores its potential. Yet, the administration’s timing, be it preoperative for enhanced immune activation or postoperative to capitalize on the surgical milieu, remains a pivotal consideration. Notably, previous studies suggest that recurrent tumor cells might not respond to BCG as robustly as they do in NMIBC [20,37]. The efficacy of intraoperative or preoperative BCG administration remains an area warranting further exploration.

The procedure of bladder cuff excision (BCE) introduces additional intricacies. Drug spillage during BCE is a genuine concern. The primary concern is drug spillage due to incomplete healing of the excision site, risking leakage into the abdominal cavity. The technical skill in suturing the BCE site is crucial. Using normal saline or distilled water minimizes leakage risks. Using normal saline or distilled water can reduce the adverse reactions associated with drug spillage into the abdominal cavity. However, this approach may concurrently elevate the risk of disseminating tumor cells, potentially leading to implantation within the peritoneal cavity. If leakage occurs, the patient, under anesthesia, may require additional sutures for reinforcement.

Given these considerations, it is clear that a singular approach may not suffice. Personalizing the strategy and factoring in patient-specific attributes, tumor pathology, and surgical specifics are of utmost importance. Advancements in this realm will undoubtedly be driven by rigorous clinical trials and a profound understanding of tumor biology, setting the stage for enhanced intraoperative bladder management following RNU.

Intraoperative instillations offer a more feasible and potentially higher utilization option [29]. As for intraoperative irrigation, while not currently a prominent area of research, it continues to be a viable strategy to reduce bladder recurrence.

## 5. Intravesical Therapy Following Radical Nephroureterectomy

Given the increased propensity for recurrence in high-risk UTUC, the standard treatment approach involves radical nephroureterectomy (RNU). Extensive investigation has been undertaken to explore the role of perioperative bladder instillation in RNU, considering both the optimal timing and frequency of instillations. This section aims to delve into the specific aspects of bladder instillation during the perioperative period of RNU, with particular emphasis on addressing these crucial questions.

### 5.1. Immediate Postoperative Single Instillation

Immediate single instillation involves administering intravesical chemotherapy directly after surgical procedures, typically within 24–48 h or even sooner [38]. Due to the multifocality and potential dissemination of tumors, residual tumor cells may still exist in the bladder after RNU surgery [9,39]. Immediate single instillation is a focused approach that specifically targets and addresses residual disease and disseminated tumor cells within the bladder, minimizing the opportunity for tumor growth [40].

It has been found in non-muscle-invasive bladder cancer that immediate single instillation following transurethral resection of bladder tumors (TURBT) helps to reduce recurrence [41]. This strategy takes advantage of the active state of tumor cells during the early postoperative period, optimizing the effectiveness of treatment [42].

Among all the studies on immediate postoperative single-dose bladder instillation, the prospective, randomized, phase II study conducted by Ito et al. holds significant representative value. Their findings demonstrated that administering a single dose of intravesical pirarubicin (THP) within 48 h after surgery significantly reduced bladder recurrence rates in UTUC patients [31]. Pirarubicin is an anthracycline anticancer drug commonly used in the treatment of various cancers. In this systematic review, two multicenter randomized clinical trials (RCT) were included. The evidence suggested that single-dose intravesical chemotherapy for UTUC patients who had undergone RNU may significantly lower the risk of bladder cancer recurrence compared to no instillation, as indicated by a hazard ratio of 0.51 (95% CI: 0.32 to 0.82, low-certainty evidence) [22]. Another meta-analysis [43], involving 532 patients from three multicenter randomized controlled trials and one large retrospective study, showed a significant reduction in bladder recurrence with an overall hazard ratio of 0.54 (95% CI: 0.38–0.76) for patients receiving intravesical instillation.

These results support the use of intravesical therapy as an effective approach to prevent bladder recurrence after RNU. Now, both EAU and AUA guidelines strongly recommend delivering postoperative bladder instillation to reduce the rate of bladder recurrence.

### 5.2. Delayed Postoperative Single Instillation

Delayed single instillation involves administering intravesical chemotherapy at a later time point after surgery, usually within one or two weeks. The delayed approach allows for proper healing of the surgical site and potentially reduces the risk of complications associated with immediate instillation, such as extravasation, the risk of which mainly depends on the suture of the bladder wall [31]. In the event of drug extravasation, it not only increases patient discomfort but also poses an increased risk of implantation [44].

In 2001, an RCT was launched to investigate the efficacy of prophylactic intravesical therapy (1 to 2 weeks after RNU) of MMC and cytosine arabinoside (Ara-C) on bladder recurrence of UTUC. In the bladder instillation group, the bladder recurrence rate was slightly lower compared to the non-instillation group, indicating a trend but not statistical significance. It should be noted that this study is relatively early, and the drug regimen and dosage are still being explored. Additionally, the sample size was small (instillation group n = 13 vs. non-instillation group n = 12), limiting the generalizability of the findings [32].

The ODMIT-C trial, which spanned six years and recruited 284 patients, demonstrated that administering intravesical mitomycin C (MMC) instillation at least one week after surgery significantly reduced the risk of bladder recurrence within one year while maintaining a low risk of complications [33]. However, Goel et al. pointed out that delaying MMC instillation for at least one week after surgery may have an impact on treatment efficacy [45].

Based on the currently available clinical evidence, the advantages of delayed instillation compared to immediate instillation are not clearly demonstrated. Furthermore, the optimal timing for delayed instillation is still under investigation and may vary depending on factors such as the specific chemotherapy agent used and individual patient characteristics.

### 5.3. Multiple Bladder Instillations

The optimal instillation regimen, whether single or multiple doses, remains a focus of investigation in the field of intravesical instillation.

In 2010, Wu et al. published a retrospective study regarding 196 UTUC patients receiving 20 mg epirubicin or 10 mg MMC six to eight times for intravesical instillation after RNU, respectively. In comparison to patients who do not undergo bladder instillation, those receiving either epirubicin or MMC exhibit reduced rates of bladder recurrence, prolonged time to bladder recurrence, and enhanced recurrence-free survival rates [22]. However, since this study did not compare single instillation with multiple instillations, no conclusion can be drawn regarding the superior efficacy of either approach.

Harraz initiated an RCT with the primary objective of comparing the effects of one-year maintenance intravesical chemotherapy (MIC) to a single intravesical instillation (SIC) in terms of reducing bladder recurrence following RNU for UTUC patients. Both groups received epirubicin 50 mg. In the MIC group, the treatment regimen involved weekly instillations for 6 weeks followed by monthly instillations for 1 year. The rates of bladder recurrence-free survival at 3, 6, and 12 months were similar between the two groups, indicating that multiple instillations of intravesical chemotherapy did not lead to a significant reduction in bladder recurrence rates [34].

For patients with in situ (CIS) bladder tumors, intravesical BCG instillation after TURBT is considered the standard treatment [46]. UTUC patients with IVR after RNU exhibited a worse prognosis compared to the primary NMIBC group, especially regarding the occurrence of secondary IVR. The results emphasize the complexities involved in managing recurrent bladder tumors in patients who have undergone RNU for UTUC [44].

Undoubtedly, multiple bladder instillations pose a greater burden to patients and exacerbate side effects. Hence, unless multiple bladder instillations exhibit compelling benefits, their widespread adoption may be challenging. As for the unsatisfying aforementioned results, current guidelines do not provide explicit recommendations for administering multiple bladder instillations in patients after RNU.

### 5.4. Bladder Instillations and Neoadjuvant Chemotherapy

Neoadjuvant chemotherapy (NAC) has become a cornerstone in managing UTUC, offering significant improvements in overall survival (OS) and progression-free survival (PFS) [47,48,49,50]. Given before surgery, NAC aims to shrink tumors and clear potential micrometastases, enhancing surgical success [51]. Recent research underscores NAC’s value, highlighting its role in reducing intravesical recurrence after nephroureterectomy for UTUC, especially in advanced cases [52].

Alongside NAC, bladder instillation acts as another adjuvant treatment, specifically targeting post-surgical bladder recurrence. While their methods differ, both therapies share a goal: improving patient outcomes and reducing disease return. NAC, given preoperatively in RNU for UTUC, exerts systemic therapeutic effects that may limit the dissemination of tumor cells. On the other hand, bladder instillation is usually applied intraoperatively or postoperatively, delivering therapeutic agents directly to the bladder, aiming to eradicate any residual tumor cells and prevent recurrence. Given the distinct administration timelines of NAC (preoperatively) and bladder instillation (intra- or postoperatively), there is minimal overlap in their therapeutic windows, ensuring that one does not impede the other’s efficacy. This temporal separation also offers a strategic advantage, potentially providing a prolonged period of therapeutic intervention against tumor cells. However, the combined strategy’s true potential remains to be fully elucidated. Future research should focus on determining the optimal sequencing of these treatments, their combined safety profile, and their overall impact on patient outcomes in UTUC. Such studies will be instrumental in refining treatment protocols and maximizing therapeutic benefits for UTUC patients.

## 6. Intravesical Therapy following Kidney-Sparing Management

Kidney-sparing surgery (KSS) is a valuable approach for managing UTUC and preserving renal function. Approaches considered for KSS were segmental ureterectomy (SU), ureteroscopy (URS), percutaneous management (PC), and chemo-ablation [53,54,55]. Different from intravesical instillation, which involves injecting chemotherapeutic agents into the bladder to prevent tumor recurrence, chemo-ablation in UTUC is a kidney-sparing technique where the chemotherapeutic agents are directly applied to the tumor site in the upper urinary tract, aiming to destroy tumor cells. The selection of the most appropriate approach depends on factors such as tumor characteristics, location, and patient-specific considerations and should be made in consultation with a multidisciplinary team of urologists and oncologists.

A systematic review and meta-analysis found no differences in oncological outcomes among different drug administration methods for UTUC or CIS of the upper urinary tract treated with KSS and adjuvant endocavitary treatment [56]. However, the efficacy of these interventions in localized low-risk UTUC had only been validated in a small-scale population for localized low-risk UTUC, as the recurrence rates following adjuvant instillations were similar to those observed in untreated patients [57]. Therefore, European Association of Urology (EAU) guidelines do not recommend postoperative bladder instillation for low-risk UTUC. According to AUA guidelines, clinicians may consider adjuvant pelvicalyceal chemotherapy and intravesical chemotherapy following UTUC ablation if no bladder or UT perforation is observed to reduce the risk of implantation metastasis (Expert Opinion) [58].

KSS can be considered for high-risk patients with imperative indications such as a solitary kidney, bilateral UTUC, chronic kidney disease, or those who are medically ineligible or unwilling to undergo RNU. However, KSS for high-risk patients may carry a higher risk of progression and reduced OS [54]. In such cases, postoperative prophylactic medication instillation is vital, and a single dose of intravesical chemotherapy is recommended to prevent recurrence [59].

Therefore, for high-risk UTUC patients who have undergone KSS and face a heightened risk of postoperative intravesical recurrence, prophylactic intravesical therapy may be considered an urgent treatment option.

## 7. Intravesical Therapy Following Ureteroscopy-Guided Biopsy

Performing pre-RNU URS biopsy aids in accurate UTUC staging and classification. This preoperative evaluation helps in surgical planning, guiding the choice between KSS and RNU [60,61]. Additionally, it serves as an effective means of screening and monitoring high-risk individuals, further emphasizing its significance in managing this patient population [62]. However, it is important to note that recent evidence suggests that URS prior to RNU has been associated with a higher risk of IVR [63].

In the first meta-analysis investigating preoperative URS prior to RNU (16 studies, n = 5489), patients who underwent URS had a significantly higher rate of bladder recurrence post-RNU compared to those without URS. However, long-term survival outcomes were comparable between the groups [64]. The findings of other meta-analyses were consistent with the aforementioned results [65,66]. Sharma et al. provided additional evidence supporting the association between preoperative URS with biopsy and increased risk of IVR after RNU, while percutaneous biopsy showed no such association [63].

These findings, along with other related studies, highlighted the need for careful consideration and close monitoring of patients who undergo URS before RNU, as it has emerged as a risk factor for postoperative IVR [67,68,69]. The increased risk of tumor dissemination during the URS procedure may account for the higher rate of bladder recurrence observed. However, URS plays a valuable role in providing accurate staging and histological diagnosis, aiding in the formulation of surgical strategies [70,71,72]. Performing preoperative URS before RNU while effectively controlling IVR incidence remains a challenge. Therefore, considering the previous discussion, postoperative bladder instillation appears to be highly effective in targeting disseminated tumor cells. The follow-up is crucial for patients who undergo both URS and RNU consecutively, as it plays a vital role in promptly diagnosing and guiding treatment for any recurrence. Timely treatment following detection is key to improving the prognosis.

Regrettably, at present, there is a lack of consensus regarding the optimal management strategy for URS prior to RNU, and a definitive standard recommendation is yet to be established. The discussion on the necessity of immediate instillation after URS has become increasingly heated, highlighting the importance of a high-quality clinical study specifically investigating the use of immediate instillation following URS procedures. There is an ongoing clinical trial (NCT05810623), but recruitment has not yet commenced.

In addition to ureteroscopy, any diagnostic interventions focused on the upper urinary tract hold the potential to elevate the incidence of post-RNU IVR. Notably, procedures such as ureteral catheterization could contribute to an increased likelihood of IVR occurrences following RNU [73].

## 8. Limitations

This comprehensive review of intravesical therapy for UTUC has certain inherent limitations that should be acknowledged. The scope of our coverage, although extensive, may not encompass all facets of intravesical therapy. This limitation is attributable to the specificity of our search keywords and criteria. We acknowledge that certain therapies from other medical domains, repurposed for UTUC treatment, might have been overlooked. Additionally, some outdated therapies that lack clinical relevance or translational potential may not be featured in this review. It is essential to recognize that even in a comprehensive review, the ever-evolving landscape of medical research can introduce new developments, making it challenging to capture every relevant aspect comprehensively. Moreover, our primary focus on clinical aspects, such as efficacy, safety, and guideline adherence, has limited the examination of crucial factors like patient preferences and economic considerations, which can significantly impact treatment decisions but are not comprehensively addressed herein. Furthermore, the interrelatedness of various components of intravesical therapy, including tumor grade and follow-up protocols, could not be exhaustively discussed due to space constraints and the need for focused analysis. Lastly, as with any review, the potential for publication bias exists, where studies with significant or positive outcomes are more likely to be published. Despite our efforts to mitigate this bias through stringent search and inclusion criteria, it is important to acknowledge this inherent limitation. As observed in the provided data, many clinical trials remain incomplete or do not publish their clinical data. Consequently, our review is limited to focusing on reported clinical trials. This inherent limitation may introduce a potential bias, as unreported or incomplete trials may yield different results. Therefore, readers should interpret our findings with an awareness of these limitations and remain vigilant for updates and emerging research in this continually evolving field.

## 9. Future Directions

Compared to the current advancements in NMIBC intravesical therapy, the field of intravesical therapy for UTUC still holds vast potential for exploration and research. It is anticipated that there will be an increased focus on optimizing the dosage, dwell time, treatment duration, and drug selection for UTUC intravesical therapy. Each component will surely be supported by a wealth of high-quality clinical evidence. Further exploration into the mechanisms (especially by identifying suitable biomarkers) might guide us in selecting the most appropriate drug regimens for UTUC patients [74,75].

The future research directions for intravesical therapy primarily focus on (1) novel instillation agents (NCT03617003 and NCT02793128) or repurposing systemic and local therapy drugs for intravesical therapy after RNU (NCT04398368 and NCT01606345); (2) combination approaches with other systemic treatments (NCT03504163); (3) standard instillation protocols following ureteroscopy with biopsy prior to RNU (NCT05810623 and NCT05731622); and so on.

Based on the current advancements, UGN-101 (MitoGel™) has shown promising results as an endocavitary administered gel-based formulation containing MMC, providing targeted treatment for urothelial carcinoma [76,77]. The OLYMPUS trial demonstrated its efficacy in treating low-grade upper tract urothelial carcinoma, with notable complete response rates and durability of response [78,79]. Moreover, UGN-101 holds the potential as a kidney-sparing treatment option for high-grade UTUC patients [80,81,82]. The efficacy of UGN-101 in treating UTUC through kidney-sparing surgery hints at its potential for future prophylactic applications against the recurrence of UTUC. An initial study aimed at its role in recurrent UTUC of the renal pelvis and ureter (NCT04006691) was withdrawn due to insufficient participant enrollment. However, just as with the original intent of this study, further research was warranted to explore its role in preventing recurrence in different parts of the urinary tract, especially intravesical recurrence.

## 10. Conclusions

In the treatment of UTUC, intravesical therapy has emerged as an important therapeutic approach, demonstrating significant progress and research outcomes. By delivering medications directly into the bladder, it reduces the risk of IVR and improves patient survival rates. Various drugs and treatment regimens have been utilized for intravesical instillation. Additionally, there is ongoing research exploring novel agents and combination therapies. However, the current clinical evidence is still relatively limited, necessitating further studies to assess and refine the efficacy and safety of intravesical therapy.

Bladder instillation for UTUC should not be viewed in isolation but rather as part of a comprehensive approach that integrates diagnosis, surgical treatment, adjuvant therapy, and follow-up. It is crucial to thoroughly study the role of intravesical therapy in specific clinical contexts. This includes the use of intraoperative bladder instillation and irrigation, which involves the administration of therapeutic agents directly into the bladder during surgery to minimize the risk of recurrence. Another significant scenario is the intravesical therapy following radical nephroureterectomy, where both single-dose and multiple-dose regimens are employed post-surgery to reduce the chances of tumor recurrence and provide sustained therapeutic effects. Furthermore, in situations where kidney preservation is prioritized, intravesical therapy post kidney-sparing procedures becomes instrumental in both managing the existing condition and thwarting the disease’s progression. It is also noteworthy that URS elevates the risk of IVR, underscoring the heightened significance of intravesical therapy in such contexts. By examining these specific scenarios, we can assess the overall impact of intravesical therapy and work towards the development of new comprehensive treatment strategies.

## Figures and Tables

**Figure 1 cancers-15-05020-f001:**
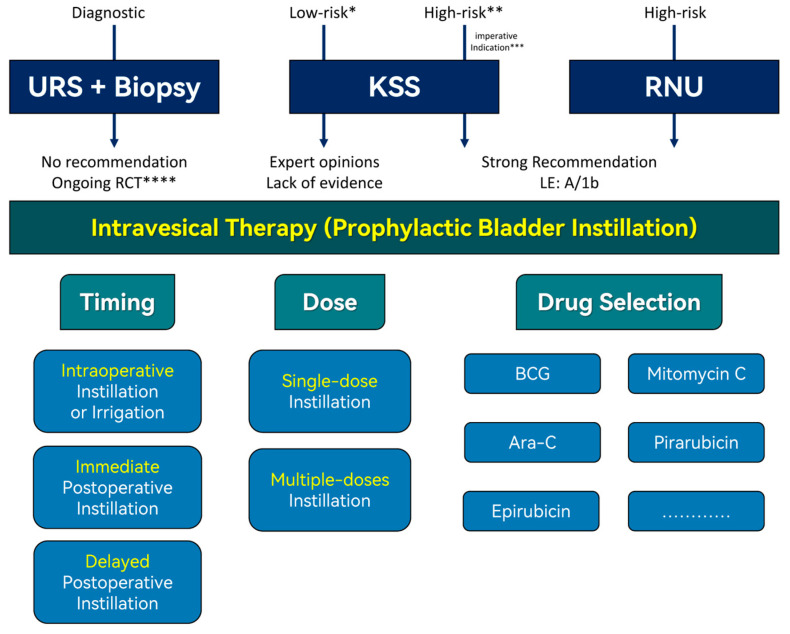
This diagram illustrates recommendations for postoperative prophylactic intravesical therapy based on clinical evidence in different clinical scenarios, as well as the timing (intraoperative instillation [28,29] or irrigation [30], immediate postoperative instillation [31] and delayed postoperative instillation [32,33]), dosage (single-dose instillation [31,33] and multiple-doses instillation [32,34]), and choice of medications for intravesical therapy. * “Low-risk” requires meeting the following conditions simultaneously: unifocal disease, tumor size < 2 cm, negative for high-grade cytology, low-grade ureteroscopy biopsy, and no invasive aspect on CT. ** “High-risk” can be satisfied by meeting any of the following criteria: multifocal disease, tumor size ≥ 2 cm, high-grade cytology, local invasion on CT, hydronephrosis, previous radical cystectomy for high-grade bladder cancer, or histological subtype. *** “Imperative indication” refers to cases involving patients with a solitary kidney, bilateral UTUC, chronic kidney disease, or patients who are medically ineligible or unwilling to undergo RNU. **** NCT05810623. URS: ureteroscopy; KSS: kidney-sparing surgery; RNU: radical nephroureterectomy; RCT: randomized controlled trial; LE: level of evidence; BCG: Bacillus Calmette–Guerin; Ara-C: cytosine arabinoside.

**Table 1 cancers-15-05020-t001:** Summary of clinical trials on intravesical therapy for prevention of intravesical recurrence after treatment for UTUC.

ID	Abbreviation	Author + Year	Status	Drugs	Dose and Timing	Follow-Up
/	/	Sakamoto 2001	Completed	MMC 20 mg and Ara-C 200 mg	A total of 28 instillations were given over 2 years.	45 months
/	/	Wu 2010	Completed	Epirubicin 20 mg or MMC 10 mg	6 to 8 times after RNU	55.6 months
ISRCTN 36343644	ODMIT-C	O’Brien 2011	Completed	MMC 40 mg	Single dose at least 1 week after RNU	12 months
/	THP Monotherapy	Ito 2012	Completed	THP 30 mg	Single dose within 48 h after RNU	24 months
UMIN000009682	/	Yamamoto 2013	Unknown status	THP	Single dose after RNU	/
NCT02438865	/	Osman 2015	Completed	Epirubicin 50 mg	Single dose with 48 h or maintenance therapy after RNU	24 months
NCT02923557	/	Li 2015	Unknown status	THP 40 mg	Single dose within 24 h after RNU	
NCT02740426	/	Li 2017	Unknown status	THP 40 mg	Single dose within 24 h after diagnostic URS	36 months
NCT03062059	/	Seo 2017	Recruiting	Gemcitabine 2000 mg	During RNU	72 months
NCT03209206	/	Ku 2017	Unknown status	Docetaxel 75 mg	Single dose within 48 h after RNU	24 months
NCT03030157	/	Huang 2019	Recruiting	THP 30 mg	Single dose within 72–168 h postoperatively plus 1 year long-term after RNU	12 months
/	REBACARE	Van Doeveren 2018	Completed	MMC 40 mg	Single dose immediately (within 3 h) before RNU or KSS	24 months
UMIN000024267	JCOG1403	Miyamoto 2018	Completed	THP 30 mg	Single dose within 24 h after RNU	36 months
NCT03658304	/	Crispen 2018	Not yet recruiting	MMC 40 mg	Single dose during RNU	36 months
NCT04398368	GEMINI	Boorjian 2020	Terminated	Gemcitabine	Single dose at least 1 h at the time of RNU	24 months
NCT05810623	MINERVA	D’Andrea 2023	Not yet recruiting	/	Single dose within 24 h after diagnostic URS	24 months
NCT05731622	SINCERE	Baard, 2023	Not yet recruiting	MMC	Single dose after URS	24 months

UTUC: upper tract urothelial carcinoma; URS: ureteroscopy; KSS: kidney-sparing surgery; RNU: radical nephroureterectomy; MMC: Mitomycin C; Ara-C: cytosine arabinoside; THP: tetrahydropyranyldoxorubicin.

## Data Availability

The data supporting this review are from previously reported studies and datasets, which have been cited.

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
