# Peer review of "Intravesical Therapy for Upper Urinary Tract Urothelial Carcinoma: A Comprehensive Review"

_cancers, 2023, doi:10.3390/cancers15205020_

Round 1

Reviewer 1 Report

Well performed overview of the different treatment options

Minor pointzs:

Shortenings like IVR, RNU must be explained at the first occurrence in the text.

Needs minor English revision.

Author Response

Dear Reviewer,

I would like to express my sincere gratitude for your constructive feedback. Your recognition of the effort put into the overview of the various treatment options is greatly appreciated.

I acknowledge the points you've raised concerning the use of abbreviations such as IVR and RNU. I want to assure you that we have thoroughly reviewed the manuscript, and we will make the necessary revisions to ensure that these abbreviations are explained at their first occurrence in the text. This will enhance the clarity of the manuscript and improve the overall reading experience for our audience.

Thank you once again for your valuable feedback, which will undoubtedly contribute to the improvement of our work.

Reviewer 2 Report

Dear Authors,

I read the paper “Intravesical Therapy for Upper Urinary Tract Urothelial Carcinoma” with interest.

The aim of the study was to comprehensively review and discuss the current status and future perspectives of intravesical therapy for upper urinary tract urothelial carcinoma (UTUC). Authors performed a good overview of current state of the art and they should be commended for their efforts.

Overall, the paper is interesting. Few remarks:

Title: Can you add something more ? (ie a review, or a comprehensive review or a not systematic review)

Abstract:adequate

Introduction: adequate.

Methods:

-       A review of the literature usually focus only on original article (eventually including case report according to the topic analyzed) and excluding other reviews (either systematic or not), editorial comment, etc. Why did you decide to include other reviews? If worthy, These studies should be cite during your discussion but not included during the article selection.

-A flowchart of extracted paper could be helpful for the readers

-       Limitations of your study should be reported in the final part of your discussion

Finally, here you can find three interesting papers about conservative management you might find useful:

-       10.23736/S2724-6051.20.03689-9

-       10.23736/S2724-6051.20.03710-8

-       10.23736/S2724-6051.22.04643-2

Quality of english seems to be good.

Author Response

We sincerely appreciate your thoughtful and thorough review of our paper, 'Intravesical Therapy for Upper Urinary Tract Urothelial Carcinoma.' Your interest in our work is greatly valued, and we're pleased to hear that you found the paper to be interesting.

We acknowledge your feedback and commendation for our efforts in comprehensively reviewing and discussing the current status and future perspectives of intravesical therapy for upper urinary tract urothelial carcinoma (UTUC).

Regarding your remarks:

  1. Title: We appreciate your suggestion, and we enhanced the title to better reflect the comprehensive nature of our review.
  2. Methods: Thank you for raising this important point. We apologize for any confusion caused. To clarify, we did not include other general reviews in our study. Instead, we incorporated the results of systematic reviews and meta-analyses as part of high-level clinical evidence in our discussion. All the clinical studies covered by these systematic reviews were individually listed and studied by us. We have made the necessary modifications based on your feedback. Thank you for bringing this to our attention!
  3. We acknowledge the limitations of our study and have detailed them in the final part of our discussion. Your recommendation to include the three papers on conservative management has been invaluable. We have carefully reviewed these articles and have incorporated relevant information, ensuring proper citation and expansion of our manuscript.

Reviewer 3 Report

This review focuses on intravesical therapy for upper tract urothelial carcinoma. The use of intravesical therapy in the setting of diagnostic ureteroscopy, nephron sparing procedures, and nephroureterectomy are discussed.

1.  Please discuss the need for intravesical therapy in patients receiving neoadjuvant chemotherapy prior to nephroureterectomy.

2.  Contemporary use of intravesical therapy in patients undergoing diagnostic ureteroscopy, nephron sparing procedures and nephroureterectomies should be reviewed.  How could utilization of intravesical chemotherapy at the time of these procedures be improved?

3.  Figure 1 list UGN-101 as an option for "prophylactic bladder instillation". To my knowledge this product is not available for this purpose currently and is not the product being studied in the use of addressing primary bladder cancer.  Additionally, to my knowledge, the product that is being studied in the bladder has not been evaluated for the indication presented by the authors.  For these reasons, it should be removed from the figure and to be included in the text of the manuscript, should be discussed in the proper context.

4. Trial NCT03658304 is missing from Table 1.  This makes me wonder what other prior and ongoing trials in this space are missing.

5. The first sentence of section 7 is misleading

6. A large part of this review is simply restating current guidelines.  I encourage the authors to provide more insightful interpretation of current data, provide clear statements on best supported practice, and challenge areas of controversy of current guidelines.

7. Section 7 should truncated significantly to summarize critical points only. 

8. Section 7 has multiple ongoing and completed trials listed by NCT numbers.  Most of these trials have little to do with the topic being reviewed.  If such trials are important enough to include in the summary section, they should be clearly discussed in the body of the manuscript. 

No major concerns

Author Response

Dear Reviewer,

First and foremost, we would like to express our sincere gratitude for your thorough and constructive feedback on our manuscript. Your insights and recommendations have been invaluable, shedding light on areas that require further refinement and enhancement.

Your candid comments have not only highlighted the areas of improvement but have also provided us with a clear direction to elevate the quality of our work. We genuinely appreciate the time and expertise you've dedicated to reviewing our manuscript, and it has been instrumental in guiding us towards producing a more robust and comprehensive paper.

We are committed to addressing the concerns you've raised and will endeavor to make the necessary revisions to enhance the overall quality and depth of our research. Below is our point-to-point response to each of the concerns you raised:

1. In response to your query about the need for intravesical therapy in patients receiving neoadjuvant chemotherapy prior to nephroureterectomy, we have extensively reviewed the literature and also borrowed insights from the combined treatment approaches in NMIBC. Both neoadjuvant chemotherapy and intravesical therapy serve as adjunctive treatments for high-grade UUTUC. However, it's worth noting that current research on this combined approach is still relatively limited.

Given the demonstrated superiority of neoadjuvant chemotherapy in enhancing overall survival (OS) and progression-free survival (PFS) in UTUC, coupled with the preventive benefits of intravesical therapy against intravesical recurrence (IVR), we believe that both treatments hold significant promise as adjunctive therapies. There is an urgent need for more comprehensive studies in this area to further elucidate their combined benefits.

In light of your valuable feedback, we have added an entire section dedicated to post-RNU instillation to further address this topic in our manuscript.

2. Thank you for highlighting the importance of discussing the contemporary use of intravesical therapy in patients undergoing diagnostic ureteroscopy, nephron sparing procedures, and nephroureterectomies.

In response to your valuable suggestion, we have dedicated a new section specifically to "intraoperative bladder instillation and irrigation." In this section, we have delved into the current protocols and have also shared our insights on the topic. Our aim is to provide a comprehensive overview that can serve as a reference for clinicians and researchers alike.

We genuinely hope that our discussion can inspire clinicians to design and conduct more robust clinical studies to validate the efficacy of various therapeutic regimens. We believe that through collective efforts and shared knowledge, we can pave the way for more effective and patient-centric treatment approaches in the future.

3. Thank you for your valuable feedback. We appreciate your keen observations. We acknowledge the point you've raised about the inclusion of UGN-101 and the potential confusion it may have caused. Your feedback is indeed well-taken, and we recognize that it didn't align with our original intent. Our intention was to discuss UGN-101 as a potential method for kidney-sparing surgery (KSS) and explore its prophylactic effects. We agree that the previous placement may have been misleading.

In light of your feedback, we have carefully reconsidered our approach. While UGN-101 may not be a current component of intravesical therapy, we recognize its potential implications for the future. As such, we have made the decision to discuss it in the 'future direction' section of the manuscript to ensure clarity and accuracy.

Once again, thank you for your valuable insights. Your feedback has been instrumental in improving the clarity and accuracy of our manuscript.

4. Thank you very much for pointing out the omission of trial NCT03658304 from Table 1. We sincerely apologize for this oversight. Upon your feedback, we revisited our sources and realized that there were indeed few trials that were previously outside our search scope. We have made efforts to include them now. Apart from the mentioned trial, there were a few others that either had not started recruitment or had unclear information. While we have removed some that were clearly duplicates, we have added the rest to ensure comprehensiveness. We deeply regret the inconvenience caused by our oversight and appreciate your attentive review. Please be assured that other key clinical trials have now been mentioned, analyzed, and listed in the table.

5. Thank you very much for your valuable feedback regarding the first sentence of section 7. We sincerely apologize for any confusion caused. Upon reflection, we recognize the discrepancy and have taken your suggestion to heart. The sentence has been revised to: 'Compared to the current advancements in NMIBC intravesical therapy, the field of intravesical therapy for UTUC still holds vast potential for exploration and research.'

6. Thank you for pointing out the emphasis on current guidelines in our review. Additionally, we have made revisions throughout the manuscript in line with your suggestions. We would like to clarify that our intention was to present well-established treatment modalities based on available clinical evidence, aiming to provide a foundation for peers in the field to further their research. While we have offered our perspectives where appropriate, the relatively limited research in this domain has led us to reference existing clinical guidelines from related areas, such as NMIBC, as a starting point for discussion. We recognize the significant differences between NMIBC and UTUC and agree that there is a pressing need for guidelines tailored specifically to UTUC.

Your insightful feedback has been invaluable, and we would greatly appreciate the opportunity to engage in further discussions with you on the treatment of UTUC. Thank you for your time and expertise.

7. Thank you for your constructive feedback regarding Section 7 of our manuscript.

In response to your comments, we have taken the following actions:

Restructuring of Section 7: We acknowledge that the previous version of this section was overly broad and lacked specificity. We have now divided it into three distinct subsections for clarity and coherence.

Limitations: As you rightly pointed out, our review has certain limitations. We are grateful for your insights, which have enabled us to enhance the quality of our work. We have now included a dedicated section that outlines these limitations, addressing both the concerns you raised and other inherent challenges we identified.

Future Direction: The trials, represented by their NCT numbers, signify the current research directions in UTUC. Our aim is to explore the future prospects of intravesical therapy for UTUC. Therefore, existing UTUC research serves as an invaluable benchmark and reference. We have discussed this in detail, with a particular focus on UGN-101, to provide a comprehensive overview of the current landscape and potential future trajectories.

Conclusion: We have separated the conclusion to stand alone, ensuring it succinctly summarizes the key findings and implications of our review, making it easier for readers to grasp the essence of our work.